# "Congruent" and "Opposite" Neurons: Sisters for Multisensory Integration and Segregation

**Wen-Hao Zhang**[1,2] *, **He Wang**[1], **K. Y. Michael Wong**[1], **Si Wu**[2]

wenhaoz@ust.hk, hwangaa@connect.ust.hk, phkywong@ust.hk, wusi@bnu.edu.cn

[1]Department of Physics, Hong Kong University of Science and Technology, Hong Kong.
[2]State Key Lab of Cognitive Neuroscience and Learning, and
IDG/McGovern Institute for Brain Research, Beijing Normal University, China.

## Abstract

Experiments reveal that in the dorsal medial superior temporal (MSTd) and the ventral intraparietal (VIP) areas, where visual and vestibular cues are integrated to infer heading direction, there are two types of neurons with roughly the same number. One is "congruent" cells, whose preferred heading directions are similar in response to visual and vestibular cues; and the other is "opposite" cells, whose preferred heading directions are nearly "opposite" (with an offset of $180°$) in response to visual vs. vestibular cues. Congruent neurons are known to be responsible for cue integration, but the computational role of opposite neurons remains largely unknown. Here, we propose that opposite neurons may serve to encode the disparity information between cues necessary for multisensory segregation. We build a computational model composed of two reciprocally coupled modules, MSTd and VIP, and each module consists of groups of congruent and opposite neurons. In the model, congruent neurons in two modules are reciprocally connected with each other in the congruent manner, whereas opposite neurons are reciprocally connected in the opposite manner. Mimicking the experimental protocol, our model reproduces the characteristics of congruent and opposite neurons, and demonstrates that in each module, the sisters of congruent and opposite neurons can jointly achieve optimal multisensory information integration and segregation. This study sheds light on our understanding of how the brain implements optimal multisensory integration and segregation concurrently in a distributed manner.

## 1 Introduction

Our brain perceives the external world with multiple sensory modalities, including vision, audition, olfaction, tactile, vestibular perception and so on. These sensory systems extract information about the environment via different physical means, and they generate complementary cues (neural representations) about external objects to the multisensory areas. Over the past years, a large volume of experimental and theoretical studies have focused on investigating how the brain integrates multiple sensory cues originated from the same object in order to perceive the object reliably in an ambiguous environment, the so-called multisensory integration. They found that the brain can integrate multiple cues optimally in a manner close to Bayesian inference, e.g., integrating visual and vestibular cues to infer heading direction [1] and so on [2–4]. Neural circuit models underlying optimal multisensory integration have been proposed, including a centralized model in which a dedicated processor receives and integrates all sensory cues [5, 6], and a decentralized model in which multiple local processors exchange cue information via reciprocal connections, so that optimal cue integration is achieved at each local processor [7].

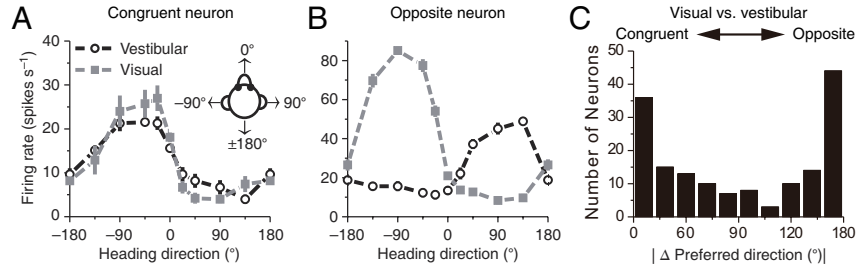

Figure 1: Congruent and opposite neurons in MSTd. Similar results were found in VIP [12]. (A-B) Tuning curves of a congruent neuron (A) and an opposite neuron (B). The preferred visual and vestibular directions are similar in (A) but are nearly opposite by $180°$ in (B). (C) The histogram of neurons according to their difference between preferred visual and vestibular directions. Congruent and opposite neurons are comparable in numbers. (A-B) adapted from [1], (C) from [13].

However, multisensory integration is only half of the story of multisensory information processing, which works well when the sensory cues are originated from the same object. In cases where the sensory cues originate from different objects, the brain should segregate, rather than integrate, the cues. In a noisy environment, however, the brain is unable to differentiate the two situations at first sight. The brain faces a "chicken vs. egg" dilemma in multisensory integration: without first integrating multiple cues to eliminate uncertainty, the brain is unable to estimate the objects reliably to differentiate whether the cues are from the same or different objects; but once the cues are integrated, the disparity information between the cues is lost, and the brain can no longer discriminate objects clearly when the cues actually come from different objects. To solve this dilemma, here we argue that the brain needs to carry out multisensory integration and segregation concurrently in the early stage of information processing, that is, a group of neurons integrates sensory cues while another group of neurons extracts the cue disparity information, and the interplay between two networks determines the final action: integration vs. segregation. Concurrent processing has the advantage of achieving rapid object perception if the cues are indeed from the same object, and avoiding information loss if the cues are from different objects. Psychophysical data tends to support this idea, which shows that the brain can still sense the difference between cues in multisensory integration [8, 9].

What are the neural substrates of the brain to implement concurrent multisensory integration and segregation? In the experiments of integrating visual and vestibular cues to infer heading direction, it was found that in the dorsal medial superior temporal area (MSTd) and the ventral intraparietal area (VIP) which primarily receive visual and vestibular cues respectively, there exist two types of neurons displaying different cue integrative behaviors [1, 10]. One of them is called "congruent" cells, since their preferred heading directions are similar in response to either a visual or a vestibular cue (Fig. 1A); and the other type is called "opposite" cells, since their preferred visual and vestibular directions are nearly "opposite" (with an offset of $180°$, half of the period of direction, Fig. 1B). Data analyses and modelling studies revealed that congruent neurons are responsible for cue integration [1, 10, 6, 7]. However, the computational role of opposite neurons remains largely unknown, despite the fact that congruent and opposite neurons are comparably numerous in MSTd and VIP (Fig. 1C). Notably, the responses of opposite neurons hardly vary when a single cue is replaced by two congruent cues (i.e., no cue integration behavior), whereas their responses increase significantly when the disparity between visual and vestibular cues increases [11], indicating that opposite neurons may serve to extract the cue disparity information necessary for multisensory segregation. Motivated by the above experimental findings, we explore how multisensory integration and segregation are concurrently implemented in a neural system via sisters of congruent and opposite cells.

## 2 Probabilistic Model of Multisensory Information Processing

In reality, because of noise, the brain estimates stimulus information relying on ambiguous cues in a probabilistic manner. Thus, we formulate multisensory information processing in the framework of probabilistic inference. The present study mainly focuses on information processing at MSTd and VIP, where visual and vestibular cues are integrated/segregated to infer heading direction. However, the main results of this work are applicable to the processing of cues of other modalities.

## 2.1 The von Mises distribution for circular variables

Because heading direction is a circular variable whose values are in range $(-\pi, \pi]$, we adopt the von Mises distribution [14] (Supplementary Information Sec. 1). Compared with the Gaussian distribution, the von Mises distribution is more suitable and also more accurate to describe the probabilistic inference of circular variables, and furthermore, it gives a clear geometrical interpretation of multisensory information processing (see below).

Suppose there are two stimuli $s_1$ and $s_2$, each of which generates a sensory cue $x_m$, for $m = 1, 2$ (visual or vestibular), independently. We call $x_m$ the direct cue of $s_m$, and $x_l$ ($l \neq m$) the indirect cue to $s_m$. Denote as $p(x_m|s_m)$ the likelihood function, whose form in von Mises distribution is

$$p(x_m|s_m) = \frac{1}{2\pi I_0(\kappa_m)} \exp\left[\kappa_m \cos(x_m - s_m)\right] \equiv \mathcal{M}(x_m - s_m, \kappa_m), \tag{1}$$

where $I_0(\kappa) = (2\pi)^{-1} \int_0^{2\pi} e^{\kappa \cos(\theta)} d\theta$ is the modified Bessel function of the first kind and order zero. $s_m$ is the mean of the von Mises distribution, i.e., the mean value of $x_m$. $\kappa_m$ is a positive number characterizing the concentration of the distribution, which is analogous to the inverse of the variance $(\sigma^{-2})$ of Gaussian distribution. In the limit of large $\kappa_m$, a von Mises distribution $\mathcal{M}[x_m - s_m, \kappa_m]$ approaches to a Gaussian distribution $\mathcal{N}[x_m - s_m, \kappa_m^{-1}]$ (SI Sec. 1.2). For small $\kappa_m$, the von Mises distribution deviates from the Gaussian one (Fig.2A).

## 2.2 Multisensory integration

We introduce first a probabilistic model of Bayes-optimal multisensory integration. Experimental data revealed that our brain integrates sensory cues optimally in a manner close to Bayesian inference [2]. Assuming that noises in different channels are independent, the posterior distribution of two stimuli can be written according to Bayes' theorem as

$$p(s_1, s_2|x_1, x_2) \propto p(x_1|s_1)p(x_2|s_2)p(s_1, s_2), \tag{2}$$

where $p(s_1, s_2)$ is the prior of the stimuli, which specifies the concurrence probability of a stimulus pair. As an example in the present study, we choose the prior to be

$$p(s_1, s_2) = \frac{1}{2\pi} \mathcal{M}(s_1 - s_2, \kappa_s) = \frac{1}{(2\pi)^2 I_0(\kappa_s)} \exp\left[\kappa_s \cos(s_1 - s_2)\right]. \tag{3}$$

This form of prior favors the tendency for two stimuli to have similar values. Such a tendency has been modeled in multisensory integration [7, 15–17]. $\kappa_s$ determines the correlation between two stimuli, i.e., how informative one cue is about the other, and it regulates the extent to which two cues should be integrated. The fully integrated case, in which the prior becomes a delta function in the limit $\kappa_s \to \infty$, has been modeled in e.g., [4, 5].

Since the results for two stimuli are exchangeable, hereafter, we will only present the result for $s_1$, unless stated specifically. Noting that $p(s_m) = p(x_m) = 1/2\pi$ are uniform distributions, the posterior distribution of $s_1$ given two cues becomes

$$p(s_1|x_1, x_2) \propto p(x_1|s_1) \int p(x_2|s_2)p(s_2|s_1)ds_2 \propto p(s_1|x_1)p(s_1|x_2). \tag{4}$$

The indirect cue $x_2$ is informative to $s_1$ via the prior $p(s_1, s_2)$. By using Eqs. (1,3) and under reasonable approximations (SI Sec. 1.4), we obtain

$$p(s_1|x_2) \propto \int p(x_2|s_2)p(s_2|s_1)ds_2 \simeq \mathcal{M}(s_1 - x_2, \kappa_{12}), \tag{5}$$

where $A(\kappa_{12}) = A(\kappa_2)A(\kappa_s)$ with $A(\kappa) \equiv \int_{-\pi}^{\pi} \cos(\theta)e^{\kappa \cos \theta} d\theta / \int_{-\pi}^{\pi} e^{\kappa \cos \theta} d\theta$.

Finally, utilizing Eqs. (1,5), Eq. (4) is written as

$$p(s_1|x_1, x_2) \propto \mathcal{M}(s_1 - x_1, \kappa_1)\mathcal{M}(s_1 - x_2, \kappa_{12}) = \mathcal{M}(s_1 - \hat{s}_1, \hat{\kappa}_1), \tag{6}$$

where the mean and concentration of the posterior given two cues are (SI Sec. 1.3)

$$\hat{s}_1 = \text{atan2}(\kappa_1 \sin x_1 + \kappa_{12} \sin x_2, \kappa_1 \cos x_1 + \kappa_{12} \cos x_2), \tag{7}$$

$$\hat{\kappa}_1 = \left[\kappa_1^2 + \kappa_{12}^2 + 2\kappa_1\kappa_{12} \cos(x_1 - x_2)\right]^{1/2}, \tag{8}$$

where `atan2` is the arctangent function of two arguments (SI Eq. S17).

Eqs. (7,8) are the results of Bayesian integration in the form of von Mises distribution, and they are the criteria for us to judge whether optimal cue integration is achieved in a neural system.

To understand these optimality criteria intuitively, it is helpful to see their equivalence of the Gaussian distribution in the limit of large $\kappa_1$, $\kappa_2$ and $\kappa_s$. Under the condition $x_1 \approx x_2$, Eq. (8) is approximated to be $\hat{\kappa}_1 \approx \kappa_1 + \kappa_{12}$ (SI Sec. 2). Since $\kappa \approx 1/\sigma^2$ when von Mises distribution is approximated as Gaussian one, Eq. (8) becomes $1/\hat{\sigma}_1^2 \approx 1/\sigma_1^2 + 1/\sigma_{12}^2$, which is the Bayesian prediction on Gaussian variance conventionally used in the literature [4]. Similarly, Eq. (7) is associated with the Bayesian prediction on the Gaussian mean [4].

### 2.3 Multisensory segregation

We introduce next the probabilistic model of multisensory segregation. Inspired by the observation in multisensory integration that the posterior of a stimulus given combined cues is the product of the posteriors given each cue (Eq.4), we propose that in multisensory segregation, the disparity $D(s_1|x_1; s_1|x_2)$ between two cues is measured by the ratio of the posteriors given each cue, that is,

$$D(s_1|x_1; s_1|x_2) \equiv p(s_1|x_1)/p(s_1|x_2), \tag{9}$$

By taking the expectation of $\log D$ over the distribution $p(s_1|x_1)$, we get the Kullback-Leibler divergence between the two posteriors given each cue. This disparity measure was also used to discriminate alternative moving directions in [18].

Interestingly, by utilizing the property of von Mises distributions and the condition $\cos(s_1 + \pi - x_2) = -\cos(s_1 - x_2)$, Eq. (9) can be rewritten as

$$D(s_1|x_1; s_1|x_2) \propto p(s_1|x_1)p(s_1 + \pi|x_2), \tag{10}$$

that is, the disparity information between two cues is proportional to the product of the posterior given the direct cue and the posterior given the indirect cue but with the stimulus value shifted by $\pi$.

By utilizing Eqs. (1,5), we obtain

$$D(s_1|x_1; s_1|x_2) \propto \mathcal{M}(s_1 - x_1, \kappa_1)\mathcal{M}(s_1 + \pi - x_2, \kappa_{12}) = \mathcal{M}\left(s_1 - \Delta\hat{s}_1, \Delta\hat{\kappa}_1\right), \tag{11}$$

where the mean and concentration of the von Mises distribution are

$$\Delta\hat{s}_1 = \text{atan2}(\kappa_1 \sin x_1 - \kappa_{12} \sin x_2, \kappa_1 \cos x_1 - \kappa_{12} \cos x_2), \tag{12}$$

$$\Delta\hat{\kappa}_1 = \left[\kappa_1^2 + \kappa_{12}^2 - 2\kappa_1\kappa_{12}\cos(x_1 - x_2)\right]^{1/2}. \tag{13}$$

The above equations are the criteria for us to judge whether the disparity information between two cues is optimally encoded in a neural system.

## 3 Geometrical Interpretation of Multisensory Information Processing

A benefit of using the von Mises distribution is that it gives us a clear geometrical interpretation of multisensory information processing. A von Mises distribution $\mathcal{M}(s - x, \kappa)$ can be interpreted as a vector in a two-dimensional space with its mean $x$ and concentration $\kappa$ representing respectively the angle and length of the vector (Fig. 2B-C). This fits well with the circular property of heading direction. When the posterior of a stimulus is interpreted as a vector, the vector length represents the confidence of inference. Interestingly, under such a geometrical interpretation, the product of two von Mises distributions equals summation of their corresponding vectors, and the ratio of two von Mises distributions equals subtraction of the two vectors. Thus, from Eq. (4), we see that multisensory integration is equivalent to vector summation, with each vector representing the posterior of the stimulus given a single cue, and from Eq. (9), multisensory segregation is equivalent to vector subtraction (see Fig. 2D).

Overall, multisensory integration and segregation transform the original two vectors, the posteriors given each cue, into two new vectors, the posterior given combined cues and the disparity between the two cues. The original two vectors can be recovered from their linear combinations. Hence, there is no information loss. The geometrical interpretation also helps us to understand multisensory information processing intuitively. For instance, if two vectors have a small intersection angle, i.e., the

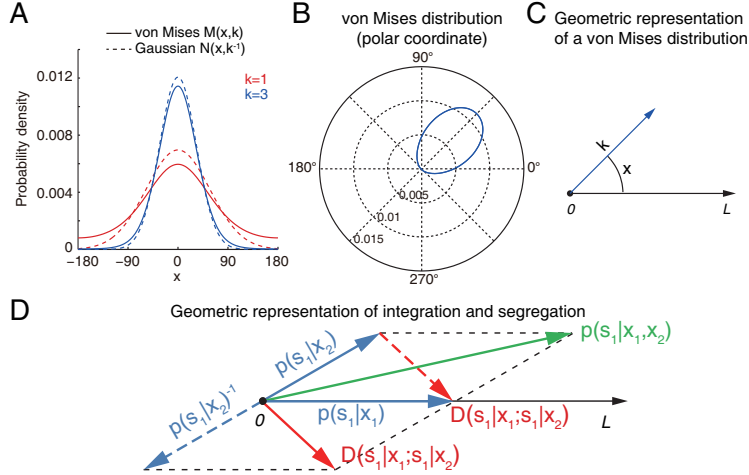

Figure 2: Geometrical interpretation of multisensory information processing in von Mises distribution. (A) The difference between von Mises and Gaussian distributions. For large concentration $\kappa$, their difference becomes small. (B) A von Mises distribution in the polar coordinate. (C) A von Mises distribution $\mathcal{M}(s - x, \kappa)$ can be represented as a vector in a 2D space with its angle given by $x$ and length by $\kappa$. (D) Geometric interpretations of multisensory integration and segregation. The posteriors of $s_1$ given each cue are represented by two vectors (blue). Inverse of a posterior corresponds to rotating it by $180°$. Multisensory integration corresponds to the summation of two vectors (green), and multisensory segregation the subtraction of two vectors (red).

posteriors given each cue tend to support each other, the length of summed vector is long, implying that the posterior of cue integration has strong confidence; and the length of subtracting vector is short, implying that the disparity between two cues is small. If the two vectors have a large intersection angle, the interpretation becomes the opposite.

# 4 Neural Implementation of Multisensory Information Processing

## 4.1 The model Structure

We adopt a decentralized architecture to model multisensory information processing in the brain [7, 19]. Compared with the centralized architecture in which a dedicated processor carries out all computations, the decentralized architecture considers a number of local processors communicating with each other via reciprocal connections, so that optimal information processing is achieved at each local processor distributively [7]. This architecture was supported by a number of experimental findings, including the involvement of multiple, rather than a single, brain areas in visual-vestibular integration [1, 10], the existence of intensive reciprocal connections between MTSd and VIP [20, 21], and the robustness of multisensory integration against the inactivation of a single module [22]. In a previous work [7], Zhang et al. studied a decentralized model for multisensory integration at MSTd and VIP, and demonstrated that optimal integration can be achieved at both areas simultaneously, agreeing with the experimental data. In their model, MSTd and VIP are congruently connected, i.e., neurons in one module are strongly connected to those having the similar preferred heading directions in the other module. This congruent connection pattern naturally gives rise to congruent neurons.

Since the number of opposite neurons is comparable with that of congruent neurons in MSTd and VIP, it is plausible that they also have a computational role. It is instructive to compare the probabilistic models of multisensory integration and segregation, i.e., Eqs. (4) and (10). They have the same form, except that in segregation the stimulus value in the posterior given the indirect cue is shifted by $\pi$. Furthermore, since congruent reciprocal connections lead to congruent neurons, we hypothesize that opposite neurons are due to opposite reciprocal connections, and their computational role is to encode the disparity information between two cues. The decentralized model for concurrent multisensory integration and segregation in MSTd and VIP is shown in Fig.3.

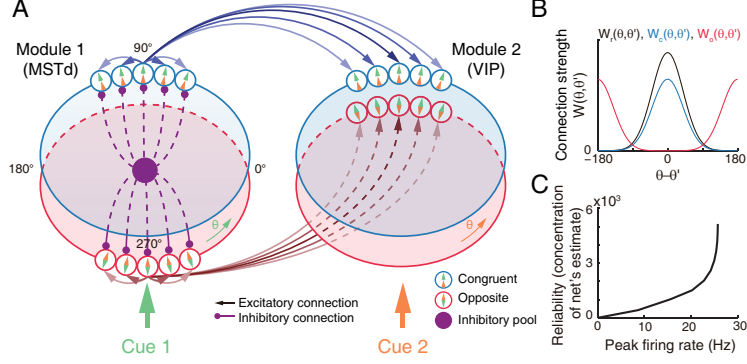

Figure 3: The model structure. (A) The model is composed of two modules, representing MSTd and VIP respectively. Each module receives the direct cue via feedforward input. In each module, there are two nets of excitatory neurons, each connected recurrently. Net $c$ (blue) consists of congruent neurons. Congruent neurons between modules are connected reciprocally in the congruent manner (blue lines). On the other hand, net $o$ (red) consists of opposite neurons, and opposite neurons between modules are connected in the opposite manner (brown lines). Moreover, to implement competition between information integration and segregation, all neurons in a module are connected to a common inhibitory neuron pool (purple, only shown in module 1). (B) The recurrent, congruent, and opposite connection patterns between neurons. (C) Network's peak firing rate reflects its estimation reliability.

## 4.2 The model dynamics

Denote as $u_{m,n}(\theta)$ and $r_{m,n}(\theta)$ respectively the synaptic input and firing rate of a $n$-type neuron in module $m$ whose preferred heading direction with respect to the direct cue $m$ is $\theta$. $n = c, o$ represents the congruent and opposite cells respectively, and $m = 1, 2$ represents respectively MSTd and VIP. For simplicity, we assume that the two modules are symmetric, and only present the dynamics of module 1.

The dynamics of a congruent neuron in module 1 is given by

$$\tau\frac{\partial u_{1,c}(\theta,t)}{\partial t} = -u_{1,c}(\theta,t) + \sum_{\theta'=-\pi}^{\pi} W_r(\theta,\theta')r_{1,c}(\theta',t) + \sum_{\theta'=-\pi}^{\pi} W_c(\theta,\theta')r_{2,c}(\theta',t) + I_{1,c}(\theta,t), \quad (14)$$

where $I_{1,c}(\theta,t)$ is the feedforward input to the neuron. $W_r(\theta,\theta')$ is the recurrent connection between neurons in the same module, which is set to be $W_r(\theta,\theta') = J_r(\sqrt{2\pi}a)^{-1}\exp\left[-(\theta-\theta')^2/(2a^2)\right]$ with periodic condition imposed, where $a$ controls the tuning width of the congruent neurons. $W_c(\theta,\theta')$ is the reciprocal connection between congruent cells in two modules, which is set to be $W_c(\theta,\theta') = J_c(\sqrt{2\pi}a)^{-1}\exp\left[-(\theta-\theta')^2/(2a^2)\right]$. The reciprocal connection strength $J_c$ controls the extent to which cues are integrated between modules and is associated with the correlation parameter $\kappa_s$ in the stimulus prior (see SI Sec. 3.3).

The dynamics of an opposite neuron in module 1 is given by

$$\tau\frac{\partial u_{1,o}(\theta,t)}{\partial t} = -u_{1,o}(\theta,t) + \sum_{\theta'=-\pi}^{\pi} W_r(\theta,\theta')r_{1,o}(\theta',t) + \sum_{\theta'=-\pi}^{\pi} W_o(\theta,\theta')r_{2,o}(\theta',t) + I_{1,o}(\theta,t). \quad (15)$$

It has the same form as that of a congruent neuron except that the pattern of reciprocal connections are given by $W_o(\theta,\theta') = J_c(\sqrt{2\pi}a)^{-1}\exp\left[-(\theta+\pi-\theta')^2/(2a^2)\right] = W_c(\theta+\pi,\theta')$, that is, opposite neurons between modules are oppositely connected by an offset of $\pi$. We choose the strength and width of the connection pattern $W_o$ to be the same as that of $W_c$. This is based on the finding that the tuning functions of congruent and opposite neurons have similar tuning width and strength [12]. Note that all connections are imposed with periodic conditions.

In the model, we include the effect of inhibitory neurons through a divisive normalization to the responses of excitatory neurons [23], given by

$$r_{1,n}(\theta,t) = \frac{1}{D_u}\left[u_{1,n}(\theta,t)\right]_+^2, \quad (16)$$

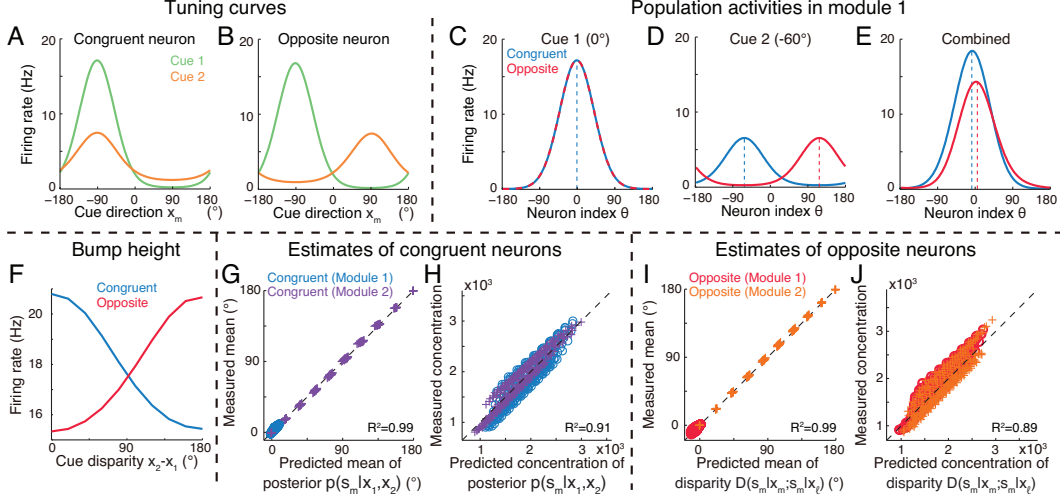

Figure 4: Bayes-optimal multisensory integration and segregation with congruent and opposite neurons. (A-B) Tuning curves of an example congruent neuron and an example opposite neuron in module 1. The preferred direction of the congruent neuron in response to two single cues are the same at $-90°$, but the preferred direction of the opposite neuron under two single cues are opposite by $180°$. (C-E) The neuronal population activities at module 1 under three cuing conditions: only the direct cue 1 (C), only the indirect cue 2 (D), and combination of the two cues (E). (F) The activity levels of the congruent and opposite neuronal networks (measured by the corresponding bump heights) vs. the cue disparity. (G-H). Comparing the mean and concentration of the stimulus posterior given two cues estimated by the congruent neuronal network with that predicted by Bayesian inference, Eqs. (7,8). Each dot is a result obtained under a parameter set. (I-J). Comparing the mean and concentration of the cue disparity information estimated by the opposite neuronal network with that predicted by probabilistic inference, Eqs. (12,13). Parameters: $J_r = 0.4\bar{J}$, $J_c = J_o \in [0.1, 0.5]J_r$, $\alpha_1 = \alpha_2 \in [0.8, 1.6]U_m^0$, $I_b = 1$, $F = 0.5$. (G-J) $x_1 = 0°$, $x_2 \in [0°, 180°]$.

where $D_u \equiv 1+\omega \sum_{n'=c,o} \sum_{\theta'=-\pi}^{\pi} [u_{1,n'}(\theta', t)]_+^2$. $[x]_+ \equiv \max(x, 0)$, and the parameter $\omega$ controls the magnitude of divisive normalization.

The feedforward input conveys the direct cue information to a module (e.g., the feedforward input to MSTd is from area MT which extracts the heading direction from optical flow), which is set to be

$$I_{1,n}(\theta, t) = \alpha_1 \exp\left[-\frac{(\theta - x_1)^2}{4a^2}\right] + \sqrt{F\alpha_1} \exp\left[-\frac{(\theta - x_1)^2}{8a^2}\right]\xi_1(\theta, t) + I_b + \sqrt{FI_b}\epsilon_{1,n}(\theta, t), \quad (17)$$

where $\alpha_1$ is the signal strength, $I_b$ the mean of background input, and $F$ the Fano factor. $\xi_1(\theta, t)$ and $\epsilon_{1,n}(\theta, t)$ are Gaussian white noises of zero mean with variance satisfying $\langle\xi_m(\theta, t)\xi_{m'}(\theta', t')\rangle = \delta_{mm'}\delta(\theta-\theta')\delta(t-t')$, $\langle\epsilon_{m,n}(\theta, t)\epsilon_{m',n'}(\theta', t')\rangle = \delta_{mm'}\delta_{nn'}\delta(\theta-\theta')\delta(t-t')$. The signal-associated noises $\xi_1(\theta, t)$ to congruent and opposite neurons are exactly the same, while the background noises $\epsilon_{1,n}(\theta, t)$ to congruent and opposite neurons are independent of each other. At the steady state, the signal drives the network state to center at the cue value $x_1$, whereas noises induce fluctuations of the network state. Since we consider multiplicative noise with a constant Fano factor, the signal strength $\alpha_m$ controls the reliability of cue $m$ [5]. The exact form of the feedforward input is not crucial, as long as it has a uni-modal shape.

## 4.3 Results

We first verify that our model reproduces the characteristics of congruent and opposite neurons. Figs. 4A&B show the tuning curves of a congruent and an opposite neuron with respect to either visual or vestibular cues, which demonstrate that neurons in our model indeed exhibit the congruent or opposite direction selectivity similar to Fig. 1.

We then investigate the mean population activities of our model under different cuing conditions. When only cue $x_1$ is applied to module 1, both the congruent and opposite neuronal networks in

module 1 receive the feedforward input and generate bumps at $x_1$ (Fig. 4C). When only cue $x_2$ is applied to module 2, the congruent neuronal network at module 1 receives a reciprocal input and generates a bump at $x_2$, whereas the opposite neuronal network receives an offset reciprocal input and generates a bump at $x_2 + \pi$ (Fig. 4D). For the indirect cue $x_2$, the neural activities it induces at module 1 is lower than that induced by the direct cue $x_1$ (Fig. 4C). When both cues are presented, the congruent neuronal network integrates the feedforward and reciprocal inputs, whereas the opposite neuronal network computes their disparity by integrating the feedforward inputs and the offset reciprocal inputs shifted by $\pi$ (Fig. 4E). The two networks compete with each other via divisive normalization. Fig. 4F shows that when the disparity between cues is small, the activity of congruent neurons is higher than that of opposite neurons. With the increase of cue disparity, the activity of the congruent neuronal network decreases, whereas the activity of the opposite neurons increases. These complementary changes in activities of congruent and opposite neurons provide a clue for other parts of the brain to evaluate whether the cues are from the same or different objects [24].

Finally, to verify whether Bayes-optimal multisensory information processing is achieved in our model, we check the validity of Eqs. (7-8) for multisensory integration $p(s_m|x_1, x_2)$ by congruent neurons in module $m$, and Eqs. (12-13) for multisensory segregation $D(s_m|x_m; s_m|x_l)$ ($l \neq m$) by opposite neurons in module $m$. Take the verification of the congruent neuronal network in module $m$ as an example. When a pair of cues are simultaneously applied, the actual mean and concentration of the networks's estimates (bump position) are measured through population vector [25] (SI Sec. 4.2). To obtain the Bayesian predictions for the network's estimate under combined cue condition (details in SI Sec. 4.3), the mean and concentration of that network's estimates under either single cue conditions are also measured, and then are substituted into Eqs. (7-8). Comparisons between the measured mean and concentration of congruent networks in two modules and the corresponding theoretical predictions are shown in Fig. 4G&H, indicating an excellent fit, where each dot is the result under a particular set of parameters. Similarly, comparisons between the measured mean and concentration of opposite networks and the theoretical predictions (SI Sec. 4.3) are shown in Fig. 4I&J, indicating opposite neurons indeed implement multisensory segregation.

## 5 Conclusion and Discussion

Over the past years, multisensory integration has received large attention in modelling studies, but the equally important issue of multisensory segregation has been rarely explored. The present study proposes that opposite neurons, which is widely observed at MSTd and VIP, encode the disparity information between sensory cues. We built a computational model composed of reciprocally coupled MSTd and VIP, and demonstrated that the characteristics of congruent and opposite cells naturally emerge from the congruent and opposite connection patterns between modules, respectively. Using the von Mises distribution, we derived the optimal criteria for integration and segregation of circular variables and found they have clear geometrical meanings: integration corresponds to vector summation while segregation corresponds to vector subtraction. We further showed that such a decentralized system can realize optimal cue integration and segregation at each module distributively. To our best knowledge, this work is the first modelling study unveiling the functional role of opposite cells. It has a far-reaching implication on multisensory information processing, that is, the brain can exploit sisters of congruent and opposite neurons to implement cue integration and segregation concurrently.

For simplicity, only perfectly congruent or perfectly opposite neurons are considered, but in reality, there are some portions of neurons whose differences of preferred visual and vestibular heading directions are in between $0°$ and $180°$ (Fig. 1C). We checked that those neurons can arise from adding noises in the reciprocal connections. As long as the distribution in Fig. 1C is peaked at $0°$ and $180°$, the model can implement concurrent integration and segregation. Also, we have only pointed out that the competition between congruent and opposite neurons provides a clue for the brain to judge whether the cues are likely to originate from the same or different objects, without exploring how the brain actually does this. These issues will be investigated in our future work.

**Acknowledgments**

This work is supported by the Research Grants Council of Hong Kong (N_HKUST606/12 and 605813) and National Basic Research Program of China (2014CB846101) and the Natural Science Foundation of China (31261160495).

## Footnotes

*Current address: Center for the Neural Basis of Cognition, Carnegie Mellon University.

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
