[Supplementary Material]

# "Congruent" and "Opposite" Neurons: Sisters for Multisensory Integration and Segregation Supplementary Information

**Wen-Hao Zhang**[1,2] * , **He Wang**[1], **K. Y. Michael Wong**[1], **Si Wu**[2]

wenhaoz@ust.hk, hwangaa@connect.ust.hk, phkywong@ust.hk, wusi@bnu.edu.cn

[1]Department of Physics, Hong Kong University of Science and Technology, Hong Kong.
[2]State Key Lab of Cognitive Neuroscience and Learning, and
IDG/McGovern Institute for Brain Research, Beijing Normal University, China.

## 1 Basics of the von Mises Distribution

### 1.1 The von Mises distribution

The von Mises probability density function for a circular variable $x$ is defined as

$$\mathcal{M}(x - \mu, \kappa) = \frac{1}{2\pi I_0(\kappa)} \exp\left[\kappa \cos(x - \mu)\right], \tag{S1}$$

where $\mu$ is the mean of $x$, and the concentration parameter $\kappa$ measures the dispersion of $x$ around its mean value. $I_0(\kappa)$ is the modified Bessel function of the first kind and zero order, which is given by

$$I_0(\kappa) = \frac{1}{2\pi} \int_0^{2\pi} \exp\left(\kappa \cos x\right) dx. \tag{S2}$$

Note that $\mathcal{M}[x - \mu + \pi, \kappa]$ equals to $\mathcal{M}[\mu, -\kappa]$. To avoid the indeterminancy of the parameter $\kappa$, it is usual to take $\kappa > 0$.

Apart from using $\kappa$ to measure the concentration, we usually use the *mean resultant length* $\rho$ to measure the dispersion of a circular variable, because it can be more easily estimated from sampled data. The mean resultant length is defined as

$$\rho = \mathbb{E}[\cos(x - \mu)]. \tag{S3}$$

Note that $0 \leq \rho \leq 1$. $\rho = 1$ means that the distribution is fully concentrated at the point $\mu$, while $\rho = 0$ means that the distribution is so scattered that there is no concentration around any particular point.

For a von Mises distribution with $\mu = 0$, its mean resultant length is calculated to be

$$\begin{aligned} \rho &\equiv A(\kappa), \\ &= \frac{1}{I_0(\kappa)} \int_0^{2\pi} \cos(x) e^{\kappa \cos x} dx. \end{aligned} \tag{S4}$$

### 1.2 Relationship to other distributions

**Relationship to the normal distribution**

When $\kappa$ is large, we let $\xi = \kappa^{1/2}(x - \mu)$, and the von Mises distribution is approximated to be

$$\mathcal{M}(\xi, \kappa) \propto \exp\left(-\kappa[1 - \cos(\kappa^{-1/2}\xi)]\right). \tag{S5}$$

Further approximating $1 - \cos(\kappa^{-1/2}\xi) = \frac{1}{2}\kappa^{-1}\xi^2 + \mathcal{O}(\kappa^{-2})$ for small $\xi$, we have

$$\mathcal{M}(\xi, \kappa) \propto \exp\left(-\xi^2/2\right) \propto \mathcal{N}[0, 1]. \tag{S6}$$

Thus, the von Mises distribution can be approximated to be a normal distribution for large $\kappa$ and small $|x - \mu|$, i.e,

$$\mathcal{M}(x - \mu, \kappa) \approx \mathcal{N}(x - \mu, \kappa^{-1}). \tag{S7}$$

**Relationship to the wrapped normal distribution**

In general, a von Mises distribution can be approximated by a wrapped normal distribution with the same mean $\mu$ and the same mean resultant length $A(\kappa)$. The wrapped normal distribution $\mathcal{WN}(x - \mu, \rho)$ is obtained by wrapping a normal distribution on a circle. For a random variable $x$, the corresponding random variable $x_w$ of the wrapped distribution is obtained by

$$x_w = x(\bmod 2\pi), \tag{S8}$$

and the wrapped distribution satisfies

$$f_w(x) = \sum_{k=-\infty}^{\infty} f(x + 2k\pi), \tag{S9}$$

where $f(x)$ is the probability density function of $x$.

Hence the probability density function of the wrapped normal distribution is defined as

$$\mathcal{WN}(x - \mu, \rho) = \frac{1}{\sqrt{2\pi}\sigma} \sum_{k=-\infty}^{\infty} \exp\left[-\frac{(x - \mu + 2k\pi)^2}{2\sigma^2}\right], \tag{S10}$$

where $\rho = \exp(-\sigma^2/2)$ is mean resultant length of the wrapped normal distribution.

By matching the mean and the mean resultant length of a von Mises distribution and a wrapped normal distribution, we have following approximation,

$$\mathcal{M}(x - \mu, \kappa) \quad \simeq \quad \mathcal{WN}(x - \mu, A(\kappa)) + \mathcal{O}(\kappa^{-1/2}), \quad \kappa \to \infty. \tag{S11}$$

It has been shown that this approximation works very well, even in the worst case when $\kappa \sim 1.4$ (ch. 3 in [1]).

### 1.3 Product of two von Mises distributions

The Bayesian integration of two cues is expressed as (see Eq. 4 in the main text which uses the fact that marginal prior is flat)

$$p(s|x_1, x_2) \propto p(s|x_1)p(s|x_2). \tag{S12}$$

Since $p(s|x_m) = \mathcal{M}[s - x_m, \kappa_m]$, for $m = 1, 2$, we need to calculate the product of two von Mises distributions, which is given by

$$p(s|x_1)p(s|x_2) = \frac{1}{(2\pi)^2 I_0(\kappa_1) I_0(\kappa_2)} \exp\left[\kappa_1 \cos(s - x_1) + \kappa_2 \cos(s - x_2)\right]. \tag{S13}$$

We have

$$
\begin{aligned}
&\kappa_1 \cos(s - x_1) + \kappa_2 \cos(s - x_2) \\
&= \kappa_1(\cos x_1 \cos s + \sin x_1 \sin s) + \kappa_2(\cos x_2 \cos s + \sin x_2 \sin s), \\
&= (\kappa_1 \cos x_1 + \kappa_2 \cos x_2)\cos s + (\kappa_1 \sin x_1 + \kappa_2 \sin x_2)\sin s, \\
&= \kappa_3 \cos(s - x_3),
\end{aligned} \tag{S14}
$$

where

$$
\begin{aligned}
\kappa_3 &= \left[(\kappa_1 \cos x_1 + \kappa_2 \cos x_2)^2 + (\kappa_1 \sin x_1 + \kappa_2 \sin x_2)^2\right]^{1/2}, \\
&= \left[\kappa_1^2 + \kappa_2^2 + 2\kappa_1\kappa_2 \cos(x_1 - x_2)\right]^{1/2}; \tag{S15}
\end{aligned}
$$

$$x_3 = \operatorname{atan2}(\kappa_1 \sin x_1 + \kappa_2 \sin x_2, \kappa_1 \cos x_1 + \kappa_2 \cos x_2), \tag{S16}$$

$$
\operatorname{atan2}(y, x) = \begin{cases}
\arctan(y/x) & x > 0 \\
\arctan(y/x) + \pi & x < 0 \text{ and } y \geq 0 \\
\arctan(y/x) - \pi & x < 0 \text{ and } y < 0 \\
\pi/2 & x = 0, y > 0 \\
-\pi/2 & x = 0, y < 0 \\
\text{undefined} & x = 0, y = 0
\end{cases} \tag{S17}
$$

After normalization, we get

$$p(s|x_1, x_2) = \frac{1}{2\pi I_0(\kappa_3)} \exp\left[\kappa_3 \cos(s - x_3)\right]. \tag{S18}$$

In the complex plane, Eqs. (S15 and S16) can be expressed as

$$\kappa_3 e^{jx_3} = \kappa_1 e^{jx_1} + \kappa_2 e^{jx_2}, \tag{S19}$$

where $\kappa e^{jx}$ denotes a vector in polar coordinates, with $\kappa$ and $x$ representing the length and angle of the vector, respectively.

### 1.4 Integral of the product of two von Mises distributions

We present how the posterior $p(s_1|x_2)$ is calculated.

$$\begin{aligned}
p(x_2|s_1) &= \int_0^{2\pi} p(x_2|s_2)p(s_2|s_1)ds_2, \\
&= \frac{1}{(2\pi)^2 I_0(\kappa_1)I_0(\kappa_s)} \int_0^{2\pi} \exp\left[\kappa_2 \cos(s_2 - x_2) + \kappa_s \cos(s_2 - s_1)\right] ds_2.
\end{aligned} \tag{S20}$$

Similar to Eqs. (S14-S16), we get

$$p(x_2|s_1) = \frac{I_0\left(\left[\kappa_2^2 + \kappa_s^2 + 2\kappa_2\kappa_s \cos(s_1 - x_2)\right]^{1/2}\right)}{2\pi I_0(\kappa_2)I_0(\kappa_s)}. \tag{S21}$$

The above equation is not a von Mises distribution, but it can be approximated as one. The two von Mises distributions in Eq. (S20) can be approximated by wrapped normal distributions, respectively (see Eq. S7), which are

$$\begin{aligned}
p(x_2|s_2) &= \mathcal{M}(x_2 - s_2, \kappa_2) &\simeq& \quad \mathcal{WN}(s_2 - x_2, A(\kappa_2)), \tag{S22} \\
p(s_2|s_1) &= \mathcal{M}(s_2 - s_1, \kappa_s) &\simeq& \quad \mathcal{WN}(s_2 - s_1, A(\kappa_s)). \tag{S23}
\end{aligned}$$

With these approximations, Eq. (S20) becomes

$$\begin{aligned}
p(x_2|s_1) &\simeq& \int_0^{2\pi} \mathcal{WN}(x_2 - s_2, A(\kappa_2))\mathcal{WN}(s_2 - s_1, A(\kappa_s))ds_2, \\
&=& \mathcal{WN}(s_2 - x_2, A(\kappa_2)) * \mathcal{WN}(s_2 + s_1, A(\kappa_s))\big|_{s_2 = 0}, \\
&=& \mathcal{WN}(x_2 - s_1, A(\kappa_2)A(\kappa_2)). \tag{S24}
\end{aligned}$$

where $*$ denotes the convolution.

Using the approximation of Eq. (S11), we finally get

$$p(x_2|s_1) \simeq \mathcal{M}\left(x_2 - s_1, A^{-1}\{A(\kappa_2)A(\kappa_s)\}\right). \tag{S25}$$

Using Bayes' theorem and the fact that $p(s_1)$ is flat, we obtain Eq. (5) in the main text.

## 2 Multisensory integration with Gaussian distribution

In the main text, we present probabilistic multisensory integration with the von Mises distribution. To see its difference with that using Gaussian distribution, we present the result for Gaussian distribution below. In the Gaussian case, the likelihood function is given by

$$p(x_m|s_m) = \mathcal{N}[x_m - s_m, \sigma_m^2] = \frac{1}{\sqrt{2\pi}\sigma_m} \exp\left[-\frac{(x_m - s_m)^2}{2\sigma_m^2}\right], \tag{S26}$$

where the inverse of the variance of Gaussian distribution is related to the concentration of von Mises distribution (Eq. 1), and $\sigma_m^{-2} \approx \kappa_m$, for large $\kappa_m$ (Eq. S7).

The stimulus prior in Gaussian distribution is written as (compared to Eq. 3),

$$p(s_1, s_2) = \frac{1}{\sqrt{2\pi}\sigma_s L_s} \exp\left[-\frac{(s_1 - s_2)^2}{2\sigma_s^2}\right], \tag{S27}$$

where $L_s = 2\pi$ for heading direction.

Substituting Eqs. (S26 and S27) into Eq. (4), the posterior $p(s_1|x_1, x_2)$ is calculated to be

$$p(s_1|x_1, x_2) = \mathcal{N}[s_1 - \hat{s}_1, \hat{\sigma}_1^2], \tag{S28}$$

where the mean and variance of the posterior are

$$\hat{\sigma}_1^{-2} = \sigma_1^{-2} + (\sigma_2^2 + \sigma_s^2)^{-1}, \tag{S29}$$

$$\hat{s}_1 = \hat{\sigma}_1^2 \left[ \sigma_1^{-2} x_1 + (\sigma_2^2 + \sigma_s^2)^{-1} x_2 \right], \tag{S30}$$

Note that the reliability of cue integration using von Mises distribution decreases with the cue disparity $(x_1 - x_2)$ (see Eq. 7), but in the Gaussian case, the reliability of cue integration $\hat{\sigma}_1^{-2}$ is independent of the cue disparity.

## 3  Theoretical analysis of the model performance

Limited by space, we only present the results of the model performance in the main text. Here, we present more detailed analysis of the model behaviors.

### 3.1  The intrinsic dynamics of a single module

We first look at the intrinsic dynamics of a single module without receiving feedforward inputs $(I_{m,n}(\theta, t) = 0)$ and reciprocal inputs from the other module $(J_c = J_o = 0)$. Under this condition, the dynamics of a single module is written as,

$$\tau \frac{\partial}{\partial t} u_{m,n}(\theta, t) = -u_{m,n}(\theta, t) + \sum_{\theta'=-\pi}^{\pi} W_r(\theta, \theta') r_{m,n}(\theta', t), \tag{S31}$$

$$r_{m,n}(\theta, t) = \frac{[u_{m,n}(\theta, t)]_+^2}{1 + \omega \sum_{n'=c,o} \sum_{\theta'=-\pi}^{\pi} [u_{m,n'}(\theta', t)]_+^2}. \tag{S32}$$

Because the recurrent connections $W_r(\theta, \theta')$ are of the Gaussian form, it can be checked that the population activities of the congruent and opposite neuronal networks in equilibrium can be approximated by the following Gaussian ansatz [2],

$$u_{m,n} \approx U_{m,n} \exp\left[ -\frac{(\theta - \hat{z}_{m,n}(t))^2}{4a^2} \right], \tag{S33}$$

$$r_{n,n} \approx R_{m,n} \exp\left[ -\frac{(\theta - \hat{z}_{m,n}(t))^2}{2a^2} \right], \quad (m = 1,\ 2;\ n = c,\ o). \tag{S34}$$

They are localized in space, called bumps, with $\hat{z}_{m,n}(t)$ being the bump position and $U_{m,n}$, $R_{m,n}$ the bump heights. Without receiving reciprocal inputs from another module, the activities of congruent and opposite neurons in the same module are completely symmetric. Thus we let $U_{m,c} = U_{m,o} = U_m$ and $R_{m,c} = R_{m,o} = R_m$ in analyzing the intrinsic dynamics of a single module.

Substituting the Gaussian ansatz into the network dynamics (Eq. S31 and S32), we obtain,

$$U_m = \frac{\rho J_r}{\sqrt{2}} R_m, \tag{S35}$$

$$R_m = \frac{U_m^2}{1 + 2\sqrt{2\pi}\omega\rho a U_m^2}, \tag{S36}$$

where $\rho = N/2\pi$ is the neuronal density with $N$ the number of congruent or opposite neurons.

Combining Eqs. (S35 and S36), it gives to

$$4\sqrt{\pi}\rho\omega a U_m^2 - \rho J_r U_m + \sqrt{2} = 0. \tag{S37}$$

The solution of the above equation is written as,

$$U_m = \frac{\rho J_r \pm \sqrt{(\rho J_r)^2 - 16\sqrt{2\pi}\rho\omega a}}{8\sqrt{\pi}\rho\omega a}. \tag{S38}$$

Thus, $U_m$ has real solutions when $J_r \geq \bar{J} = 4(2\pi)^{1/4}(\omega a/\rho)^{1/2}$. $\bar{J}$ is the minimal strength for the network to hold an active state in the absence of external input. Since no persistent activity is reliably observed in multisensory brain areas, we choose $J_r$ to be smaller than $\bar{J}$ in our model.

## 3.2 The dynamics of reciprocally connected modules

We then analyze the dynamics of reciprocally connected modules. In response to noisy feedforward and reciprocal inputs, the bump positions of congruent and opposite neurons, $\hat{z}_{m,n}$, fluctuate over time, and their means and variances can be analyzed theoretically. As an example, we consider the bump positions of congruent neurons in both modules ($\hat{z}_{1,c}$ and $\hat{z}_{2,c}$). The dynamics of opposite neurons can be similarly analyzed.

The bump activity of congruent neurons in module 1, $u_{1,c}$, is influenced by the activities of another two groups of neurons: the activity of opposite neurons in the same module, $u_{1,o}$, and the activity of congruent neurons in the other module, $u_{2,c}$. From Eq. (S32), we see that $u_{1,o}$ only influence the height, rather than the position, of $u_{1,c}$ via divisive normalization. On the other hand, the reciprocal inputs from $u_{2,c}$ will affect the bump position of $u_{1,c}$. Therefore, in computing $\hat{z}_{1,c}$, we only need to consider the influence from $\hat{z}_{2,c}$, and the effect of $u_{1,o}$ is included in the bump height of $U_{1,c}$ implicitly.

We can project the high-dimensional network dynamics onto its dominating modes to simplify the network dynamics significantly. For the network with translation-invariant connections, its dominating dynamical mode is the displacement mode [2], which, in our case, is written as

$$\phi_1(\theta|\hat{z}_{m,c}) = \left(\frac{\theta - \hat{z}_{m,c}}{a}\right) \exp\left[-\frac{(\theta - \hat{z}_{m,c})^2}{4a^2}\right], \tag{S39}$$

where $\hat{z}_{m,c}$ denotes the bump position of congruent neurons in module $m$, and $a$ the bump width.

When two cues are close enough ($|x_1 - x_2| \ll a$), we substitute the above Gaussian ansatz into the network dynamics (Eqs. 14, 15, 16), and then project them onto the displacement mode (Eq. S39). Projecting a function $f(\theta)$ onto a motion mode $\phi_1(\theta|\hat{z})$ is to compute the quantity $\int f(\theta)\phi_1(\theta|\hat{z})d\theta / \int \phi_1(\theta|\hat{z})^2 d\theta$. After projection, we obtain the dynamics of the bump positions of congruent neurons in two modules, which are

$$\frac{d\hat{z}_{1,c}}{dt} = g_{12}(\hat{z}_{2,c} - \hat{z}_{1,c}) + h_1(x_1 - \hat{z}_{1,c}) + \beta_1\eta_1(t), \tag{S40}$$

$$\frac{d\hat{z}_{2,c}}{dt} = g_{21}(\hat{z}_{1,c} - \hat{z}_{2,c}) + h_2(x_2 - \hat{z}_{2,c}) + \beta_2\eta_2(t), \tag{S41}$$

where $\langle\eta_m(t)\eta_{m'}(t')\rangle = \delta_{mm'}\delta(t - t')$. And the effective reciprocal connection strengths $g_{ml}$, the effective feedforward input strengths $h_m$, and the effective noise strengths $\beta_m$ are given by

$$g_{ml} = \frac{\rho J_c R_{l,c}}{\sqrt{2}\tau U_{m,c}}, \quad h_m = \frac{\alpha_m}{\tau U_{m,c}}, \quad \beta_m^2 = \frac{4aF}{\sqrt{2\pi}(\tau U_{m,c})^2}\left[(2/3)^{3/2}\alpha_m + I_b\right]. \tag{S42}$$

The parameters $U_{m,c}$ and $R_{m,c}$ are the means of the bump heights of $u_{m,c}$ and $r_{m,c}$ in equilibrium.

The mean and variance of $\hat{z}_{m,c}$ in equilibrium can be analytically solved as,

$$\langle\hat{z}_{1,c}\rangle = \frac{(g_{21} + h_2)h_1 x_1 + g_{12}h_2 x_2}{g_{12}h_2 + g_{21}h_1 + h_1 h_2}, \tag{S43}$$

$$V(\hat{z}_{1,c}) = \frac{[(g_{21} + h_2)\text{tr}(\mathbf{M}) + g_{12}g_{21}]\beta_1^2 - g_{12}^2\beta_2^2}{2\text{tr}(\mathbf{M})(g_{12}h_2 + g_{21}h_1 + h_1 h_2)}, \tag{S44}$$

where $\text{tr}(\mathbf{M}) = -(g_{12} + g_{21} + h_1 + h_2)$.

## 3.3 The cue integration performance of the model

For the convenience of analysis, we consider that all parameters of two congruent neuron networks are symmetric and set $g_{12} = g_{21} = g$, $h_1 = h_2 = h$, $\beta_1 = \beta_2 = \beta$ in Eqs. (S43 and S44). Under this simplification, the mean and variance of the bump position of congruent neurons in module 1 are written as

$$\langle \hat{z}_{1,c} \rangle = \frac{(g^{-1} + h^{-1})x_1 + h^{-1}x_2}{2h^{-1} + g^{-1}}, \tag{S45}$$

$$V(\hat{z}_{1,c})^{-1} = \frac{2}{\beta^2}\left[h + (g^{-1} + h^{-1})^{-1}\right]. \tag{S46}$$

Comparing the above results with the Bayesian predictions (Eqs. S29-S30), and considering $\sigma_1 = \sigma_2 \equiv \sigma$, we get the below relationship,

$$h \propto \sigma^{-2}, \tag{S47}$$

$$g \propto \sigma_s^{-2}, \tag{S48}$$

which states that the cue reliability $\sigma^{-2}$ is encoded in the effective feedforward input strength $h$ to a module, and that the variance of prior $\sigma_s^{-2}$ is encoded in the effective reciprocal connection strength $g$ between modules.

# 4 Simulation of the model performance

Here, we introduce the detail about simulation experiments we carried out to get the performance of the model.

## 4.1 Model parameters

In the simulation, each type of network (congruent or opposite) in a module consisted of $180$ neurons, which were uniformly distributed in the range of $(-180°, 180°]$. The two modulesp were symmetric, i.e., all of the structural parameters were the same, except that they received different cues and independent noises. The synaptic time constant $\tau$ was rescaled to 1 as a dimensionless number, and the time step size was $0.01\tau$. All connections had the same width, i.e., $a = 40°$.

We list the typical values for adjustable parameters used in simulation if not mentioned otherwise. The recurrent connection strength $J_r = 0.4\bar{J}$, where $\bar{J} = 4(2\pi)^{1/4}(\omega a/\rho)^{1/2} = 1.26$ is the minimal strength for holding persistent activity without feedforward inputs. Thus, no persistent activity occurred in each network after withdrawing the feedforward inputs. The strength of the reciprocal connections $J_r = J_o$ are in the range of $[0.2, 0.6]J_r$ and are always smaller than the recurrent connections. The input strength $\alpha_1 = \alpha_2$ was scaled relative to $U_m^0 = J_c/8a\omega\sqrt{\pi} = 8.93$ and distributed in the region of $[0.7, 1.5]U_m^0$, where $U_m^0$ is the synaptic bump height that a network can hold without feedforward input when $J_r = \bar{J}$. The interval of the input strength was in the sub-additive region of neural response curve, because sub-additive response was widely observed in experiments [3]. The strength of the background input was $I_b = 1$, and all Fano factors $F$ of the cues and background inputs were set to $0.5$. This resulted in a Fano factor of single neuron responses in the order of 1. In the simulation, the activity bump position was estimated by using a population vector (see below). Specific parameter settings are mentioned in the figure captions.

## 4.2 Model performances

For $n$-type neurons in module $m$, we used population vector to read out its estimate $z_{m,n}$ of the stimulus value from the population activity $r_{m,n}$, that is,

$$z_{m,n} = \text{Angle}\left(\sum_{\theta} r_{m,n}(\theta)e^{j\theta}\right), \tag{S49}$$

where $j$ is the imaginary unit, the operation $\text{Angle}(\cdot)$ outputs the angle (in radian) of a vector.

We measured the model performances under three cuing conditions: only cue 1, only cue 2, and both of them. In each cuing condition, the corresponding feedforward input was applied, and we read out

the estimates of the congruent and opposite neuronal networks by population vector. This process was repeated for many trials, and we fit the distribution of estimates by the von Mises distribution. Denote $z_{m,n}(i|x_l)$ as the bump position in $i$-th trial when only cue $x_l$ ($l = 1, 2$) was presented. The mean and concentration of $z_{m,n}(i|x_l)$ averaged over trials are calculated to be,

$$\langle z_{m,n}|x_l \rangle = \text{Angle} \left( \frac{1}{N} \sum_i e^{jz_{m,n}(i|x_l)} \right), \tag{S50}$$

$$\kappa(z_{m,n}|x_l) = A^{-1} \left\{ \left| \frac{1}{N} \sum_i e^{jz_{m,n}(i|x_l)} \right| \right\}, \tag{S51}$$

where $N$ is number of trials, and $A^{-1}(\cdot)$ is the inverse of the function $A(\cdot)$ given in Eq. (S4). The results for combined cues, $\langle z_{m,n}|x_1, x_2 \rangle$ and $\kappa(z_{m,n}|x_l, x_2)$, can be similarly calculated.

### 4.3 Theoretical predictions

**For congruent neurons**

The theoretical prediction for optimal multisensory integration is given by Eqs. (7-8) in the main text. From the relations between Eqs. (S15-S16) and Eq. (S19), the posterior of the stimulus $s_1$ given combined cues can be expressed in a more concise form as

$$\hat{\kappa}_1 e^{j\hat{s}_1} = \kappa_1 e^{jx_1} + \kappa_{12} e^{jx_2}. \tag{S52}$$

We measured the performances of congruent neurons given single cues, and calculated the theoretical prediction based on the above equation, which gives

$$\tilde{\kappa}_{m,c} e^{j\tilde{z}_{m,c}} = \sum_{l=1}^{2} \kappa(z_{m,c}|x_l) e^{j\langle z_{m,c}|x_l \rangle}, \tag{S53}$$

where $\tilde{z}_{m,c}$ and $\tilde{\kappa}_{m,c}$ denote, respectively, the predicted mean and concentration of the posterior of $s_m$ in multisensory integration.

**For opposite neurons**

The optimal multisensory segregation for stimulus $s_1$ (Eqs. 12-13) can be also written in a concise form, which is,

$$\begin{aligned} \Delta\hat{\kappa}_1 e^{j\Delta\hat{s}_1} &= \kappa_1 e^{jx_1} - \kappa_{12} e^{jx_2}, \\ &= \kappa_1 e^{jx_1} + \kappa_{12} e^{j(x_2+\pi)}. \end{aligned} \tag{S54}$$

In network implementation, the position of cue 2 (indirect cue) encoded by opposite neurons in module 1 has been rotated by $\pi$ due to the opposite reciprocal connections between opposite neurons in two modules. As a result, the Bayesian predictions for the mean and concentration of opposite neurons' estimates under combined cue conditions are

$$\tilde{\kappa}_{m,o} e^{j\tilde{z}_{m,o}} = \sum_{l=1}^{2} \kappa(z_{m,o}|x_l) e^{j\langle z_{m,o}|x_l \rangle}. \tag{S55}$$

where $\tilde{z}_{m,o}$ and $\tilde{\kappa}_{m,o}$ denote, respectively, the predicted mean and concentration of the disparity information of $s_m$ in multisensory segregation.

## Footnotes

* Current address: Center for the Neural Basis of Cognition, Carnegie Mellon University.