[Reviews · NeurIPS 2016]

Reviewer 1

Summary

Here the authors present a Bayesian model of cue combination for circular variables and a new notion of “disparity,” which they argue may allow multisensory segregation in parallel with typical multisensory integration. They present results from probabilistic modeling (with a geometric interpretation) and neural networks. The main purpose of the paper is to describe a functional role for “opposite” cells in MSTd and VIP, and this is well achieved on all levels.

Qualitative Assessment

It would be interesting to understand how the disparity model relates to other models of multisensory integration/segregation. E.g. the Kording et al. 2007 model of causal inference is very different and would not require two populations. Some justification for the flat prior p(s_m) seems warranted. I can easily imagine that the distribution of natural head movements is not uniform. Additionally, it is unclear how priors play a role in disparity. Although the paper is clearly written and presents some interesting results, it's a bit difficult to see whether this approach will have a broad applicability. My (maybe naive) understanding of previous models (Deneve/Pouget) was that cues were generally represented in separate sensory areas and then a posterior might be represented in downstream multisensory areas. To what extent these types of congruent/opposite tuning curves exist in other systems is unclear.

Confidence in this Review

2-Confident (read it all; understood it all reasonably well)


Reviewer 2

Summary

The authors identify two types of neurons in MSTd and VIP: congruent and opposite, coding for two sensory variables. Congruent neurons have the same tuning, while opposite neurons have (ideally) 180 degree difference in preferred variable. While the congruent neurons have been shown to help with cue integration, the role of opposite neurons is not well known. The authors note that the brain needs to do both cue integration and segregation, and suggest the opposite neurons help with the latter. In their view, the congruent and opposite populations are their to simultaneously estimate the integration and segregation. To test this hypothesis the authors make a simple neural network with two modules, representing MSTd and VIP, with congruent and opposite neurons in each module. The modules are connected to each other, with contgruent neurons being connected congruently (so no phase difference), and opposite neurons oppositely (with a 180 degree phase difference). It is thanks to these weights that the congruent and opposite neurons when measured actually act according to their name. Given this network, the authors show that, indeed, the congruent and opposite populations perform integration and segregation respectively, and the this is also done optimally compared to a bayesian model (as derived from von Mised probability distributions).

Qualitative Assessment

The overal model and reasoning was clearly explained, and within the proposed model I completely agree with the authors about the function of congruent vs opposite neurons and like their approach. However, I believe the results are currently over-interpreted in the sense that it is not clear how this actually relates to the brain yet (an example of over-interpretation is lines 235-236. ): First, the proposed weight structure in the model is well explained, but poorly justified biologically. Are opposite and congruent neurons indeed connected close to how it is proposed in this paper? This currently seems more like a prediction than an assumption. (Although would I be right in saying that broadly the same results could be gotten by a non-dynamical model where one just assumes passive tuning of neurons? I.e. such that no assumption about the underlying network structure etc is needed.) Secondly, while the authors claim a 'small portion' (line 271) of neurons actually are neither fully congruent nor opposite, a rough count of Fig. 1C suggests actually over half of the neurons are in between. I think this suggests something more is going on than just two populations doing separate tasks. In any case, this is currently only briefly discussed (how would the authors expect a more fuzzy connectivity to change the results?). Additionally, I did not see any discussion on the effect of different parameter sets (besides that different parameter sets were used for Figures 4G-J), nor was it entirely clear exactly how the parameters were varied. While I appreciated the thorough explanation of von Mises functions, I think this could be shortened quite a bit to allow for more discussion to implement the above comments. The paragraph about the von Mises function in the discussion, in my opinion, does not add much new to the paper or the field in general and can be left out as well (that the von mises function is a useful tool in neuroscience has been established for a long time). Clarity and presentation: The figures were very clear and smooth, and not much effort was needed to understand them. (I would give the figures a score of 4 in fact) The discussion felt more rushed than the rest, and was harder to parse. I would suggest a run through of the grammar and sentence structure.

Confidence in this Review

2-Confident (read it all; understood it all reasonably well)


Reviewer 3

Summary

Authors have proposed a biologically-plausible mechanism that segregates multisensory cues in parallel to cues integration. Although their idea is that, in general, integration of multisensory information has to be somehow linked to segregation in parallel, authors use one popular example: the sense of direction of self-motion, or heading direction. There are two important neural groups, one of them known to play an important role in information integration. Authors introduce a network model that poses the second group as responsible for information segregation. Authors also propose a connectivity pattern for these networks.

Qualitative Assessment

Overall, authors have proposed an interesting role for the opposite group of neurons. The modeling seems interesting, and the problem is certainly appealing for biology and applications. The general idea of combining segregation and integration of multisensory information is appealing, and is definitely the strongest point of the paper. The text is sometimes not as clear as possible, and could benefit from a proof reading. I have also some comments on the modeling, but my main concern would be on a lack of closer connection to biology and real data. Below, I will expand on these points. Regarding writing, maybe the first point that really needs addressing is defining what “heading direction” is: I have failed finding it in the text. I believe authors meant by heading direction as the notion of direction in the self-motion of animals, as being vastly studied in animals such as primates. Although I think is not strictly necessary or mandatory, I would try and define the term right in the abstract. However, in the introduction the term should be defined as early as possible. This keyword is not something easily recognizable by someone who may have not heard of the term before, and this could generate confusion. The text needs proof reading for improving its clarity and readability. Not grammatically, but the phrasing could be improved. For instance, in many sentences “the” was used instead of “a” (as in “we introduce first the probabilistic model of Bayes-optimal multisensory integration). In some sentences it could be omitted (“The psychophysical data tends to support…”). Another example in this direction is qualifying connections among neurons from the congruent group as congruent, as from the opposite group as opposite. It only becomes clear when figure 3 is shown, where congruent (opposite) neurons from an angle alpha connect to neurons that respond to angle alpha (alpha + 180o). In summary, I do not believe “congruent” and “opposite” are clearly defining these two patterns. Especially because this is one of the results of this paper, improving this description would likely increase the paper impact on readers. Authors claim that “the brain is unable to differentiate the two situations straightforwardly.” Is there any psychological or psychophysical experiment showing that? Would the authors elaborate on that? This problem seems very similar to the “Cocktail Party Problem”, and it is believed that many brains excel on that. Can authors elaborate on the sentence (51) “… shows that when multisensory integration happens, the brain can still sense the difference between cues” and how exactly this relates to the previous sentence on line (47)? Is this related to effects such as ventriloquism (as mentioned in ref 8), where visual cue comes from one object, but auditory cue comes from another? Maybe this could be clarified in the text. Definition of s1, s2 and x1,x2 are not clear yet for me. Especially, the way these two variables are used in equation (1) indicate some loss of generality that might not have been stated. Is it the case? Maybe a brief sentence with an example would be very helpful in line (82). When describing the integration model, in line 94 authors assume that the noise from different (sensory) channels are independent. Is there any experimental guarantee on this matter? As an example, during motion would not auditory and visual cues might be correlated by Doppler effect? Except for equation 16, there might be a definition missing regarding the firing rate r for both m groups. Finally, the results show interesting and promising qualities regarding the model proposed in this paper, but there is a lack of comparison with what is observed. Are there any experimental results that can be contrasted with the results? Are there any public datasets that could be used with this current model? Is there any prediction exclusively from this model that could be tested experimentally?

Confidence in this Review

2-Confident (read it all; understood it all reasonably well)


Reviewer 4

Summary

Authors investigate a well-defined problem in sensory integration (or segregation): They investigate how certain neural populations serve as a bayesian machinery for sensory segregation (which is not investigated/experimented much in neuroscience field). They approach the problem both in the algorithmic level (see the problem as bayes-optimal inference) and both implementation (biophysical modelling).

Qualitative Assessment

I have the following suggestions for the authors: 1/ Authors might be able to shorten the introduction by less explanation on multi-sensory integration and segregation (i.e. line 22-52). I think some high level explanation is enough to convey their message (and let the reader flow faster) 2/ Author probably could motivate better the need for probabilistic models for "multi-sensory information processing" in the brain. I recommend them [just as an example] to have a look at this article: http://www.ncbi.nlm.nih.gov/pubmed/17895984 3/ I'd suggest to change the title of section 2 to "Backgrounds" as a more appropriate title 4/ I'd suggest not to use the notation "atan2"(Eq. 7); I didn't see to be used before 5/ At line 162 might have to refer to an appropriate reference. 6/ Typo line 170: Neuron --> neuron (after ",") 7/ I didn't find this sentence conniving "Since the number of opposite neurons is comparable with that of congruent neurons in MSTd and VIP, it is plausible that they also have a computational role." I didn't understand how does the comparability of the population size imply the excistance of any computational role? 8/ The "discussion and conclusion" section, to me, mainly is a summary of the paper. One potential thingy to discuss there is elaboration on the connection of bayesian inference and a biologically plausible implementation of it in the brain (which authors tackle its core question nicely). For example the what is the potential source of the noise which "induce fluctuations of the network state"? I would recommend to have a look at this papers: 8a- http://www.ncbi.nlm.nih.gov/pubmed/21212356 8b- http://www.sciencedirect.com/science/article/pii/S0896627314001044 8c- http://www.jneurosci.org/content/36/5/1775.short

Confidence in this Review

1-Less confident (might not have understood significant parts)


Reviewer 5

Summary

It is thought that in MSTd and VIP neural responses to visual and vestibular stimulation are integrated as part of the computation to estimate heading direction. In recent years, two relatively distinct types of neurons have been reported in both visual areas. Both types have their response level modulated by visual and vestibular input, but in different ways. Congruent neurons prefer inputs that maximally agree. Opposite neurons prefer inputs that maximally disagree. The manuscript aims to develop a computational account of the functional purpose of the opposite neurons. They suggest that opposite neurons may play an important role in segregating multimodal input. This hypothesis is not new. But the authors undertake a serious attempt to construct a computational framework that makes rigorous the hypothesis. I can quibble with certain aspects of their model. But overall, I think it is a very nice contribution.

Qualitative Assessment

The most significant concern I had with the manuscript was that all the modeling results were presented with a fano factor of 0. The authors took pains to develop the model so that they could take into account the effects of pseudo-realistic neural noise, but they made no attempt to justify why their presentation of results did not include it. There was not even any discussion of why the authors made this seemingly odd choice. Do results change markedly as the fano factor is manipulated? Why or why not? Are the results sensitive to the level of noise? Is there a regime in which results are qualitatively unaffected by the level of noise (as one would hope would be true in a biological system)? Or are they brittle? Abstract: “there exist two types of neurons with comparable numbers.” Hard to parse. “with comparable numbers”. Please rephrase. Abstract: “the sisters of congruent and opposite neurons can jointly achieve optimal multisensory information integration and segregation” Confusing. What are ‘sisters’ of congruent and opposite neurons. Line 22: “Our brain senses the external world with multiple sensory modalities, including vision, audition…” This comment is epistemological in nature, but the brain does not sense the external world. The sense organs sense proximal stimuli. The transduced proximal stimuli are then processed into estimates or representations of the external world. The authors are surely aware of this distinction but I think it is important to be clear about it in written text. There is a lot of loose language in science and the media about these issues. We practitioners should be precise. Line 32: “The neural circuit models underlying optimal multisensory integration were also proposed…” The neural circuit models? Maybe- ‘Neural circuit models underlying optimal multisensory integration have been proposed’ Line 39: “In an ambiguous environment, however, the brain is unable to differentiate the two situations straightforwardly” Nonsense sentence. If input stimuli are ambiguous there is no correct answer, so it cannot be straightforward to differentiate one situation from another. Please rephrase so that the point is clear. Not sure what the authors are going for here. Line 41: “brain is unable to perceive…” Brains don’t perceive (or decide or realize or love). Animals and people do. Line 51: “The psychophysical data tends to support this idea, which shows that when multisensory integration happens, the brain can still sense the difference between cues [8].” The authors might consider citing Girshick et al 2009 here. Line 53: “…brain to realize…” Line 61: “modelling" Spelling. Line 71: “In reality, because of noises, the brain…” Unorthodox sentence. And noisES? Should it be ‘noise’? Line 77- “The stimulus we consider is heading direction, which is a circular variable with values in the range of (􀀀_; _] , we therefore adopt the von Mises distribution (Supplementary Information Sec. 1).” Ungramattical sentence. Line 106: “The indirect cue x2 is informative to s1 via the prior p(s1; s2) . By using Eqs. (1,3) and under reasonable approximations (SI Sec. 1.4), we obtain [posterior apprx. proportional to Von Mises distribution]” The authors should probably cite Murray & Morgenstern (2010). Their paper directly examines Bayesian cue integration of circular variables under a von Mises assumption. Line 169: “In their model, MSTd and VIP are congruently connected, i.e., Neurons in one module are strongly connected to those in the other module having the similar preferred heading directions.” Grammar. Figure 4 Caption: “but the preferred direction of the opposite neuron under either cues are opposite by 180_” Grammar. Line 222: “We then consider the population activities of our model under different cuing conditions without input noise (F = 0 )” Motivate this. Why do the authors consider the response properties with a fano factor of 0, rather than a more realistic value? Potentially confusing language. The authors refer to a fano factor as ‘input noise’. But standard usage would call the parameter, which characterizes the ratio of a neuron’s response variance to its mean, as affecting ‘output noise’ rather than ‘input noise’. Line 252: “Over the past years, multisensory integration has received large attention in modelling studies,” Grammar. Line 253: “The present study proposes that opposite neurons, which has been widely observed atMSTd and VIP but with origin and computational role unresolved, serve to encode the disparity information between sensory cues.” Hard sentence to parse. Line 259: “To our best knowledge, this work is the first modelling study unveiling the functional role of opposite cells.” Line 260: “It has a far-reaching implication on multisensory information processing, that is, the brain can exploit sisters of congruent and opposite neurons to implement cue integration and segregation concurrently.” Grammar. And what are sisters? Appendix Line 35: “The Bayesian integration of two cues is expressed as…” Please be more precise with the language. The equation stated is only correct under a flat prior. The authors write this later in the appendix but it would be could to state it here. Appendix Line 55: “von pMises” Typo. Appendix Line 143: “The strength of the background input was Ib = 1 , and all Fano factors F of the cues and background inputs were set to 0.5 . This resulted in a Fano factor of single neuron responses in the order of 1. In the simulation, the activity bump position was estimated by using a population vector (see below).” Please discuss this issue in more depth. It is not clear why different units should have different fano factors. Appendix Line 171: “perceived by opposite neurons” Neurons don’t perceive.

Confidence in this Review

3-Expert (read the paper in detail, know the area, quite certain of my opinion)